# Prospect of DFT Utilization in Polymer-Graphene Composites for Electromagnetic Interference Shielding Application: A Review

**DOI:** 10.3390/polym14040704

**Published:** 2022-02-11

**Authors:** Jonathan Tersur Orasugh, Suprakash Sinha Ray

**Affiliations:** 1Department of Chemical Sciences, University of Johannesburg, Doorfontein, Johannesburg 2028, South Africa; jsurth@gmail.com; 2Centre for Nanostructures and Advanced Materials, DSI-CSIR Nanotechnology Innovation Centre, Council for Scientific and Industrial Research, Pretoria 0001, South Africa

**Keywords:** density functional theory, solid-state chemistry techniques, EMI shielding, graphene, nanocomposite, composite, polymer

## Abstract

The improvement in current materials science has prompted a developing need to capture the peculiarities that determine the properties of materials and how they are processed on an atomistic level. Quantum mechanics laws control the interface among atoms and electrons; thus, exact and proficient techniques for fixing the major quantum-mechanical conditions for complex many-particle, many-electron frameworks should be created. Density functional theory (DFT) marks an unequivocal advance in these endeavours. DFT has had a rapid influence on quintessential and industrial research during the last decade. The DFT system describes periodic structural systems of 2D or 3D electronics with the utilization of Bloch’s theorem in the direction of Kohn–Sham wavefunctions for the significant facilitation of these schemes. This article introduces and discusses the infinite systems modelling approach required for graphene-based polymer composites or their hybrids. Aiming to understand electronic structure computations as per physics, the impressions of band structures and atomic structure envisioned along with orbital predicted density states are beneficial. Convergence facets coupled with the basic functions number and the *k*-points number are necessary to explain for every physicochemical characteristic in these materials. Proper utilization of DFT in graphene-based polymer composites for materials in EMI SE presents the potential of taking this niche to unprecedented heights within the next decades. The application of this system in graphene-based composites by researchers, along with their performance, is reviewed.

## 1. Introduction

Within recent decades, computer-based simulations founded upon the quantum-mechanical portrayal of the relationships between atomic nuclei along with its electrons has presented an increasing number of vital effects on material chemistry/physics and other scientific niches, currently not only in critical thought but additionally aimed at technological innovations [1,2,3,4,5,6,7,8,9,10]. These simulative modeling’s are carried out through an atomistic element with the aid of fixing the Schrödinger mathematical statement towards obtaining forces as well as energies, requiring just the inputted elements’ atomic numbers, as well as describing the inter-atomic bonding with high precision. Schrödinger’s mathematical statement for intricate many-electron, many-atom systems is analytically non-solvable, in addition to having numerical approaches valuable to chemistry, physics, as well as material sciences. These computational attempts were achieved in 1964 when Walter Kohn et al. established the DFT system, an idea primarily centered upon the density of the electron, being the characteristic of solely 3D (three-dimensional) coordinates [1]. The DFT mathematical statement of Kohn–Sham equations throws the problematic complexity of the electron–electron interfaces into a high-quality individual-particle feasible estimated using exchange-correlation functional ($E_{xc}\left[n\right]$) [1]. The $E_{xc}\left[n\right]$ describes the intricate kinetic as well as lively interfaces of a given electron in conjunction with another electron. Even though the structure of $E_{xc}\left[n\right]$ which rebuilds the many-body Schrödinger model is undetermined/not known, approximated functionals have been shown to define many fabric characteristics.

Effective algorithms developed for resolving Kohn–Sham mathematical models are nowadays carried out in an increasing number of erudite codes, enhancing DFT technique usage. Innovative doorways are opening in groundbreaking research in materials throughout chemistry, physics, materials science, nanoscience/nanotechnology, and other niches like biology and earth sciences. This improvement is not academically constrained, seeing as the DFT approach also uses software in many specific industrial research niches. Progress in DFT usage is so rapid that many modern purposes could not have been accomplished three years ago or considered five years earlier.

Evident improvement in current materials chemistry/physics resulted in increasing demand for apprehending the observable fact defining the characteristics and processes of the material atomistically [11,12,13]. Quantum mechanics laws regulate the atoms and electrons interfaces; thus, precise and effective approaches aimed at resolving the fundamental quantum-mechanical mathematical models for intricate many-electron, many-atom schemes ought to be created. The DFT technique inscribes a significant step forward in this regard [14], as well as within the previous decades, seeing it has had a swiftly developing influence on prototypical utilization and also industrial research.

The concept of the electronic structure was founded upon 3D periodic objects among other primary systems. The associate electronic band structural conception scopes came in the decades after quantum physics was discovered and initial instances may be seen within Sommerfeld. The nuclear physicist (1933), Isopod (1934) et al. nearly half a century ago found the overall electrons’ energy associated nuclei in a crystalline solid basic to be a focus of theoretical studies: summing up across the reciprocal region (k-space) is needed. Hence, in the Seventies, various articles concerning k-space sampling were revealed [2,3,4,5,6,7]. Given that computational strength may be restricted, the accentuation was to keep the quantity of *k*-points to be treated as low as conceivable, and the most effective decision of uncommon *k*-points was the major subject of these prior works.

Meanwhile, this data found its approach into textbooks. For instance, a fascinated reader might check it out in chapter four of Martin (2004) [8]. Considering the rise of pc power, systematic physicists, and materials researchers began to operate upon supplementary and more advanced systems comprising many atoms in one unit cell. Since such a giant unit goes together with a small Brillouin zone (BZ) in a reciprocal area, sampling of k-space received less attention. Moreover, liquid or amorphous systems that lack translational order have been approximated by giant supercells, e.g., using quasi-random structures [9]. Due to the shortage of ’true’ physical periodicity, calculations with giant supercells typically use simply one k-point that, for further procedural saving, is usually chosen to be the Ŵ-point, i.e., the origin in the reciprocal area. As has become apparent, only a few papers regarding k-point sampling appeared around the turn of the century. In recent times, performing calculations with correct results has revived concern with enhanced strategies towards reciprocal area sampling. The major steering issue is born of materials science computation. With respect to natural philosophy, like diagrams calculations, extremely amalgamated overall bulk material elementary unit cell energies are necessary [10,11]. The necessary calculations ought to be performed in an instinctive approach, mistreating the approaches of high-output computation. Hence, the use of mechanically generated densely packed k-point sets permitting the achievement of overall energy precision is better than 1 meV for each atom adopted. In a contemporary investigation [12], aimed at ensuring the targeted accuracy at all phases, a 5000 *k*-points/Å^−3^ k-point density is usually essential. Likewise, primarily machine learning-based approaches are used for the k-point grids selection that is most fitted for the matter at hand. One other innovation driving issue within the k-point sampling niche is that the curiosity in bulk materials’ unique properties, especially within electronic transport, magnetism as well as topological material states niches, has caused adaptively enhanced systems. At the same time, the overall energy may be an amount whose variation is with regard to small deviations within the charge density (CD), and so is computationally strong, that its utilization is cited higher than needing the resolution of a satisfactory architecture within the Brillouin zone so as to get a correct description of the properties of interest.

This review projects the necessity of DFT utilization in polymer–graphene composites systems design and fabrication for EMI shielding applications. Although we tend to limit ourselves to DFT calculations during this review, a first periodic systems treatment employing quantum chemistry wavefunction strategies is another possible result of any errors arising from density functional approximation. Lately, the DFT technique has drawn attention, reportedly attaining vital development [13]. Assertions on the subject of DFT computations in this article are based on approximations by Hartree–Fock, which, similar to DFT, illustrates the wavefunction with regard to single-particle orbitals. Economical pc codes for Hartree–Fock estimations comprising a choice to handle the entire schemes are obtainable. Post-Hartree–Fock strategies integrate electronic links in varied estimated aspects; instances pertinent to periodic procedures include the coupled-cluster technique or Møller–Plesset perturbation theory [13]. The complexity related to handling relationships between two electrons in different solids’ unit cells ought to be resolved. The finest ground-state correlated wavefunction might have a smaller amount of symmetries compared to Hamiltonians’ many-particle; thus, methodologies utilizing translational crystal symmetry should be considered cautiously. A scientific approach towards embodying electronic relationships calculations in periodic structures can be obtained via the increments strategy [14]. The quantum Monte Carlo technique is an alternate possibility appropriate for periodic approaches [15,16], that permits an additional versatile mathematical illustration of the many-particle wavefunction in comparison to strategies ranging from atomic orbitals’ BS. At the same time, we have a tendency to lecture amateurs in DFT niche calculations, thereby providing an outline pertaining to essential information regarding DFT periodic systems computations that have been for years amassed though the spread in literature within the publications that are not suitable for beginners. Meanwhile, the Kohn–Shams’ technique’s necessities are delineated within a range of literature [8,17,18,19], and here we show a tendency to presume readers are at home with the basics. Convergence problems relating to plane-wave growth of Kohn–Sham wavefunctions [20] or other atom-cantered BS [18,21] ought to be resolved by DFT practitioners prior to changing their interest to periodic systems; once more, we have a tendency to see the articles. 

In particular, with reference to other niches innovative utilization of DFT for novel graphene-based polymeric materials as per available literature, the key element of originality of the present article is to have highlighted the prospect of DFT utilization in the design/engineering of novel graphene-based polymeric materials for EMI shielding application and other technological advancements.

In order to develop and assess DFT techniques, a finite basis set is typically adopted. DFT codes utilize plane-waves and/or atomic-orbital form basis sets (BS) in DFT estimations. On the whole, the problem of the dependency of DFT calculations on a basis set is ignored by many users of DFT: this should not be done. The influence of basis sets on the outcome of DFT calculations has been reported [22]. In general, double and/or triple-zeta quality basis sets are adopted in DFT studies. However, it is not initially plain that basis sets optimized for wave-function ab initio approaches are the best alternative for DFT [22]. We know that there are newly developed basic sets, but these are applied for specified functionals and not a general case. The basis set chosen influences the final results obtained. 

A group of authors in their studies observed that, with respect to the diverse properties scrutinized, such as energies as well as gradients, the Pople basis sets are suggested [22]. They observed the fitting of Dunning’s basis sets with CISD, resulting in considerably higher errors in spite of possessing a larger basis functions amount: however, yielding higher error compared to 6-3111G(3d f,2pd) BS with respect to energies as well as gradients, the TZ2P BS gave the most inclusive error for general gradient approximation (GGA) functionals with the inclusion of the ZMP exchange-correlation potentials in the fitting. The error due to the BS, at the triple-zeta level that could still be substantial, reveals that the functionals acquired through fitting to a single BS stand exchangeable with other BSs.

Thus, it is most likely not critical to arrive at the premise put forth line when growing new density functionals since the generally DFT error is more significant. BS created for DFT techniques may lighten this issue; however, the issue remains regarding which functional is used [22]. A similar examination for mixture density functionals shows that the measure of careful exchange depends on the BS itself. Although BS of two-fold zeta quality gave minima ~28%, the triple-zeta BS assessed had their minima ~18%. With regards to the contrast between the mixture ‘‘BS functionals’’, similar ends with respect to the GGA functionals can be reached. Similar patterns are additionally apparent while assessing a few other communicated functionals in an enormous test set. We can find that few hybridized functionals, such as B97-1 and the t-HCTH hybridized functional present errors, are altogether lower than those acquired by B3LYP. As per an unadulterated GGA functional, the HCTH functional sorts present errors similar to B3LYP for the examined properties. In any case, the precision of current DFT could not measure up to that of ab initio extrapolation techniques [22]. It is established by Boese et al. that the preeminent functional tried yields an RMS energy error as extensive as 6.3 kcal/mol for a huge arrangement of atoms/molecules, which is as yet distant from the ideal ‘‘chemical precision’’ of 1–2 kcal/mol [22].

## 2. Crystallography Basics

The periodic orientation of atoms in crystal form is mathematically delineated by its least periodic component, the component, as well as the lattice of points invariant below transformations. The *R* lattice points fulfill the expression:(1)$$R=n_{1}a_{1}+n_{1}a_{2}+n_{3}a_{3}$$

Thus, *n*_1_; *n*_2_; *n*_3_ represent (+ve or −ve) integers. *a*_1_, *a*_2_, *a*_3_ lattice vectors traverse the 3D building block with volume Ω. The building block could also be occupied by one or many atomic entities; within the last case, the crystallographer’s decision is that of the locations of the atomic entities inside the building block on the (crystallographic) basis. The building blocks potential shapes are a measure restricted by the concerns that the unit cell periodic recurrences should be space-filling, meaning there are not any voids/overlaps. The delineated lattices with Equation (1) satisfy these conditions referred to as Bravais lattices. 

Considering some lattice space, it should normally be the case that separate pattern symmetries shaped with entire atoms (along with crystallographic basis defined atoms) are harmonious with the permanence underneath conversions specified with Bravais lattice (the topmost level in hierarchical cataloging), with the exclusion of unit cells of, for example, icosahedra and/or dodecahedra. Nevertheless, it should encompass lesser symmetry compared to a Bravais lattice. In fact, this ends up in a better crystal structures classification [23].

Considered among the mathematical niche for the group theory is the crystal symmetry, where the relevant groups (defined as purpose teams and space units) consist of a finite range of N group symmetry operations. Concerning a given cluster, its crystallographic teams should be shut underneath the insertion of any combined or amalgamated operations, wherever the combinations suggest that the sequent use of two separate symmetry operations (= cluster components, α = 1, ... N group) after each other. Underneath the term crystallographic purpose group, one addresses a precise assortment of separate symmetry operations, like reflections or rotations, in the computed logic which maps (no less than) the single purpose of the crystal lattice (which is taken into account as infinite in this regard) onto itself, whereas the other lattice purpose could also be epitomized against a dissimilar lattice purpose. The idea of the purpose cluster will not develop with respect to transformations; if we have a tendency to need the crystal, in addition to permanence underneath the purpose cluster processes, to conform to translational symmetries underneath nearly symmetrical processes TRn, as we attain the (richer) idea of a Bravais lattice. The whole Bravais lattices (BL) comprising an equivalent set of separate symmetries, i.e., having the same purpose cluster, are said to belong to an equivalent crystal system. An example is the blocky crystal system that contains the straightforward cubic, body-centered blocky (bcc) and face-cantered blocky BL. The purpose cluster, however, could also be reduced to a subgroup if the idea is less symmetrical than the BL itself. The overall range of purpose teams is thus higher or equal to the amount of BL. The conceptual cluster symmetry processes actions on associate degree electronic wave perform *ψ*(*r*) could be delineated by real-space vectors algebra, for instance,
(2)$$T_{R_{n}}\psi =\psi \left(r+R_{n}\right)$$

Conversely, the *r* electron position is moved or altered by the *R_n_* lattice vector. With regards to rotation/reflection, the operational symmetry is denoted by the matrix *Gα*, like,
(3)$$G_{\alpha }\psi \left(r\right)=\psi \left(G_{\alpha }r\right)$$

In the event of a crystal possessing a specified symmetry, the equivalent operational symmetry working on the wavefunction alters it by zero across a part problem,
(4)$$T_{R_{n}}\psi \left(r\right)=e^{i\emptyset_{n}}\psi \left(r\right)$$
(5)$$G_{\alpha }\psi \left(r\right)=e^{i\emptyset_{n}}\psi \left(r\right)$$

For rational and irrational numbers *ψ_n_* and *ψ_α_* the operational symmetry cited to this point, also known as symmorphic operational symmetry, consists of renditions, reflections, as well as rotations that have the property in common such that every solitary one of them exits the crystal (to be unbounded as well as infinite) indifferently. We will expect instances wherever the crystal is left indifferent solely via a particular symmorphic operational symmetry amalgamation. Two instances of these doubtful non-symmorphic symmetry operations are area units that glide the plane. The concerned crystal remains indifferent exclusively beneath an amalgamated reflection along with a translation, usually via a segment of the occupied lattice vector as well as the secured axis. The crystal remains indifferent exclusively under an amalgamated revolution as well as translation, usually via a tiny proportion of the comprehensive lattice vector. Through the existence or non-existence of these non-symmorphic symmetries, the categorization theme for the crystals is created by even a lot of various others instead of the purpose teams solely. Thus, the symmetries of the crystal, as well as the non-symmorphic symmetries, should eventually be delineated via the crystallographical dynasty teams. The predominant significance of the quantum mechanics symmetry is eminent. With respect to its applicability to crystals, it entails that the crystals’ Hamiltonian attenuates together with all cluster point parts. Throughout this perspective, the cluster parts area unit is described by operators upon the Hilbert space. Accordingly, the Hamiltons’ operator eigenfunctions have specific characteristics in relation to the appliance of operational symmetry.

### 2.1. Bloch’s Theorem

This theory specifically allows us to think about translational operations symmetry [15]. Even though the circumstances resulting in Bloch’s theorem will be evident in many-particle schemes via the introduction of a man-made ’simulation cell’ Hamiltonian [15,16], here, our issues are confined to single-particle eigenfunctions for individual electrons. Considering the crystals’ translational permanence, the crystals’ electronic wave functions will amend solely up to a part issue beneath translation,
(6)$$\psi \left(r+R_{n}\right)=e^{ikR_{n}}\psi \left(r\right)$$

One other corresponding mathematic approach for affirming this prerequisite is that the wave function is mandated to comprise of a lattice-periodic factor u(r), identical in every unit cell; u(r+R)=u(r), augmented by the plane wave,
(7)$$\psi_{k}\left(r\right)=e^{ikr}u_{k}\left(r\right)$$

k index refers to a vector, it might be deemed as ‘quantum number’ illustrating periodic crystals’ wave functions. K signify the momentum of the crystal. N.B: $u_{k}\left(r\right)\;$ is dependent upon k index, notwithstanding the reliance is weak in the majority of instances. Reciprocal R lattice on behalf of each assumed lattice point collection, R may be constructed traversed with the R vectors $b_{1}$, $b_{2}$, $b_{3}$, described as:(8)$$a_{i}b_{j}=2\pi \delta_{ij}$$

In reality, the $b_{i}$ (in 3D area) is acquired in the form:(9)$$b_{i}=2\pi \frac{a_{2}\times a_{3}}{det\left(a_{1}a_{2}a_{3}\right)}$$

The two R vectors formulae for the extra ones could be acquired via cyclic indices (*i*, *j*, *k*) = (1, 2, 3) permutations.

The vol. of the factual-space unit cell, Ω = |$det\left(a_{1}a_{2}a_{3}\right)$| is contained in the denominator. The R cell unit is also referred to as the Brillouin zone (BZ), such as in Ashcroft and Mermin, 1976. Owing to Bloch’s theorem as represented in Equation (7), it should be understood that the wave functions $\psi_{k}$ for the momentum of the crystal k inside the elementary cells section of the R, that is, inside the 1st BZ. If a wavefunction is translationally acted upon, the momentum of its crystal remains unchanged. The reflection or rotational activities of $G_{\alpha }$ upon the crystals’ wave-function, characterized by a real space matrix exponentiation $G_{\alpha }$, runs through its inverse in the reciprocal area, that is, it charts $k7\to G-1_{\alpha k}$.

In DFT, the whole energy of associate degree atoms, molecules, or their collection is acquired via the tallying of influences from all its electrons. Owing to the effectual single-particle portrayal presented in the Kohn–Sham mathematical model coupled with Bloch’s proposition, the whole crystal energy is often computed from the $\psi_{k}\left(r\right)$ (Equation (7)) data by taking into consideration the electronic states wherever r changes within a single unit real space, often a simplification computationally. Nonetheless, Bloch’s proposition requires the calculation of the wavefunctions for the entire boundary conditions indicated in Equation (6) using *k*-points contained by the primary BZ. With regard to a finite crystal, the quantity of terms stipulated for amalgamation across equals its unit cells quantity matching the sufficient test group, as we habitually express the BZ integration seeing the substantial total and the integral are measured interchangeably. So far, nothing would be gained in terms of procedural savings. Nonetheless, considering the fact that the wave function lattice-periodic half uk(r), is usually solely weak depending upon k, it is possible to sample the Brillouin zone integral at a finite, typically rather than a tiny number of, points. The numerical techniques to realize this square measure is based on Fourier construction. For physicalness, let us take into account some lattice-periodic perform F(k) that will comprise implicit dependencies upon the eigenvalues εi(k) and wavefunctions ψk(r). We assume that these performances are often distended into the Fourier elements finite number, up to some *Rm* = (*xm*, *ym*, *zm*),
(10)$$F\left(k\right)=\overset{m}{\underset{j=1}{\sum }}F_{j}e^{ikR_{j}}$$

F(k) integral across the occupied BZ is specified by the Fourier bottom element, $F_{0}$, the integral is meant to be estimated with the finite add across the k-points. Then, the subsequent concerns regarding error discretization suppose the enormous Fourier elements (beyond $R_{m}$) disappearance, and these are applied during a stringent instance for insulators or semiconductors (SCDs) solely. With reference to metals, Fermi surface presence wherever the bands change occupation rapidly apply from 0 to at least 1 signifying the availability of high-level Fourier elements in F(k). With a target to reduce the error arising from data discretization, k-points are introduced. Within the plainest instance of an isometric crystal having a lattice constant a, homogeneous use of equal k-points grids that occupy the BZ is recommended. E.g., in an instance where the grid comprises of Nx-points within the x-direction, hence, the finite augment as per Equation (10) is in a position to denote the integral specifically, for Nxa≥xm, for $R_{m}$ primary part. Conversely, the finite addition is necessary for the integral if the operated F(k) remained adequately fast and contained solely Fourier parts that reached dead set Nx lattice constants in real space. In the instance of non-cubic-unit-cells, the k-points might remain selective to be aligned/placed on parallel planes to the spanned planes with dual R vectors, meaning, analogous to the surfaces of the reciprocal unit cell. For a crystal possessing purpose symmetries, the symmetry processes will be wont to scale back the quantity of k-points that associate factual computations of the Kohn-Sham wave functions must be carried out. Conversely, it is appropriate to experiment solely with the non-reducible BZ wedge, which ends up in significant savings of machine prices. By virtue of symmetry mapping, one will continuously ’unroll’ the irreducible wedge to recover the total BZ. In this way, it is continuously possible to recover the absolutely crucial CD, forces, etc., required. Although the unit has no point cluster symmetries, time-reversal changelessness permits to map the wavefunction ψk to ψ−k=ψ*k, and so the quantity of k-points will be declined by an element of 2. This approach, utilizing time-reversal permanence, is fragmented using an external field or magnetization, and it does not apply to spin–orbit interfaces in a system that is short of symmetry inversion. Hence, symmetries exploitation is highly recommended.

DFT-derived codes being utilized additionally symmetrize the CD as well as atoms generated forces as the case may be. The generated results are of importance: the symmetry could not be explored if we desire to examine relaxations arising from symmetry-breakage of the atoms beginning from associate “perfect” cruciate assembly, during this instance, the majority of the codes offer am associate choice to manually throw away the symmetrization (with the consequence of considerably increased machine required resources). Hence, these forces do not seem to be symmetrized along with the numerical misestimation is decent to move the scheme from a cruciate initial point to the symmetry-breaking phase. 

In early process work, once memory devices were the main restraint, the main emphasis was on associate degree economic alternative of k-points for BZ sampling which created its potential to perform computations for a real-space unit of cells having several atomic entities.

With reference to the plainest instance, we may determine the BZ integral using the integrand, referred to as the Baldereschi function [4]. However, aimed at precise computations, a bigger set of specialized k-points are recommended. The criteria for convergence, furthermore, as express listings of specialized k-points set for 3D crystal, are reportedly resolved [5]. We must begin with procedure symmetrized functions Am(k), which are measured as the distinctive of an exact “shell” of k trajectories. m index is related to the length |k|, per specific shell having the identical *k*-points around the $\hat{W}$-point, marking the beginning of the lattice reciprocal.

### 2.2. Kohn–Sham Theorem

The expression, ψ(r(N)) of the Schrödingers’ many-body expression used for the quantum theory of N interrelating electrons in an external potential, v(r), relies on the overall coordinates which are consequently tough to unravel. The Kohn–Sham scheme [23] was acquainted with in concurrence with DFT [24] being a fictional scheme of non-intermingling particulate matter producing identical densities like the actual ground-state challenge. Conventional Kohn–Sham model presents the energy of the system as a function of the electron density, n(r), is presented by Equation (11):(11)$$E\left[n\right]=\int v\left(r\right)n\left(r\right)dr+T_{s}\left(n\right)+U_{H}\left[n\right]+E_{xc}\left[n\right]$$
(12)$$T_{s}\left[n\right]=\langle \Phi |{Ŷ}^{N}|\Phi \rangle$$

*E_xc_[n]* symbolizes the exchange-correlation functional. The K.E (kinetic energy) of the system of non-interrelating particulate matter is denoted as in the next equation.
Where Φ (*r*_1_, ..., *r_n_*) = Φ (*r*_(*N*)_) is a single Slater determinant built outside N bottom energy solutions, $f_{j}\left(r\right)$, of the Kohn–Sham expressions 12, 13, and 14, respectively:(13)$$\left[-\frac{1}{2}\nabla_{r}^{2}+v_{KS}\left(r\right)\right]f_{j}\left(r\right)=\epsilon_{j}f_{j}\left(r\right)$$
and
(14)$$U_{H}=\frac{1}{2}\iint \frac{n\left(r_{1}\right)n\left(r_{2}\right)}{\left|r_{1}-r_{2}\right|}dr_{1}dr_{2}$$
is the classical expression of Hartree energy for Coulomb energy (C.E); while the density is expressed as:(15)$$n\left(r_{1}\right)=\int {\left|\Phi (r_{\left(N\right)})\right|}^{2}dr_{2}\ldots ..dr_{N}=\overset{N}{\underset{j=1}{\sum }}{\left|f_{j}\left(r_{1}\right)\right|}^{2}$$

The systems’ ground state is determined via the minimization of the entire energy functional (as in Equation (1)). As per Hohenberg and Kohn [24], the elementary adjustable density functional theory is its density, and its ground state form that the energy remains fixed in connected with variations in the density, *δE* [*n*]/*δn* = 0. From this state and *δT_s_* /*δn* = − *v*_*KS*_, the Kohn–Sham potential appearance in Equation (13) must gratify;
(16)$$v_{KS}\left(r\right)=v\left(r\right)+\frac{\delta U_{H}\left[n\right]}{\delta n\left(r\right)}+\frac{\delta E_{xc}\left[n\right]}{\delta n\left(r\right)}$$

As previously highlighted in the main article by Hohenberg and Kohn [23], the standard model, $U_{H}\left[n\right]$, denoting the C.E has been long applied for convenience, permitting a facile and effectual way of accomplishing the C.E functional derivative relating to its density. The ease, nevertheless, comes at a precipitous price: as revealed above, the usage of this term acquaints with self-interaction effects, the non-physical interface given electron by itself. Once the density (as in Equation (5)) is adopted in the Hartree term, it yields in the numerator inside the integral terms of the form, *f_j_*(*r*_1_), *f_j_*(*r*_2_), unfolding the interaction of two entities/particles within the same state. Adopting the Hartree term in regard to a singular-electron system evidently reveals the SI problem, including a particle in an assumed state interacting with itself in that state, explicitly in practice not remunerated by the $E_{xc}\left[n\right]$, which was well-defined to call off the SI error precisely. Nevertheless, this function is unfamiliar and, in reality, substituted by means of some approximate expression. Several enactments in regard to diverse estimated schemes for *E_xc_*[*n*] have assisted in revealing the true state of matters: approximate terms can be premeditated to produce satisfactory approximations to recognized results on a case-by-case basis, even in extensive modules of systems and properties; nonetheless, none can be revealed prior to placate the fundamental prerequisite of the second Hohenberg–Kohn theorem, that is that any estimated handling of the energy functional ought to produce a higher bound to the precise value of the ground form energy in the full expression of the equivalent Schrodinger equation. Let us say the adoption of an assumed $E_{xc}\left[n\right]$ in the Kohn-Sham DFT could lead to a lower energy in comparison to the ground state energy in the many-body equation of the corresponding Hamiltonian of the concerned theory, given that Kohn–Sham formalism is not utilized.

#### Kohn–Sham Equations Reformulation

Quantum mechanically precise representation of the C.E is stated with regards to the density pair, *n*(*r*_1_, *r*_2_), instead of its product(s):(17)$$U^{QM}=\iint \frac{n\left(r_{1},\;\;\;r_{2}\right)}{\left|r_{1}-r_{2}\right|}dr_{1}dr_{2}$$

Here, the density pair is gotten by integrating the anti-symmetric, N-particle wave function, $\Psi (r_{N})$, through the entire coordinates:(18)$$n(r_{1},\;r_{2})=\frac{N\left(N-1\right)}{2}\int {\left|\Psi (r_{N})\right|}^{2}dr_{3}\ldots dr_{N}$$

Evidently, the equivalent C.E is identity-interfacing free. With regard to the Kohn–Sham theorem, defined using single Slater determining factor, the non-intermingling density pair (signified as s subscript) is expressed thus:(19)$$n_{s}\left(r_{1},\;r_{2}\right)=\frac{1}{4}\sum_{ij}\delta_{\sigma_{i}\sigma_{j}}{\left|f_{i}^{\sigma i}\left(r_{1}\right)f_{j}^{\sigma j}\left(r_{2}\right)-f_{j}^{\sigma j}\left(r_{1}\right)f_{i}^{\sigma i}\left(r_{2}\right)\right|}^{2}$$
where the *f_i_^σi^* (*r*) represents filled states/orbitals: that is, the N least energy equations in Kohn–Sham solutions, while *s_i_* denotes the spin index of the precise orbital. The non-interfacing pair of density could break into two fragments; where its role in the Hartree classical term as well as the exchange portion, $J_{s}$ is represented thus:(20)$$n_{s}\left(r_{1},\;r_{2}\right)=\frac{n\left(r_{1}\right)n\left(r_{2}\right)+J_{s}\left(r_{1},\;r_{2}\right)}{2}$$
(21)$$J_{s}\left(r_{1},\;r_{2}\right)=-\underset{ij}{\sum }\left[f_{i}^{\sigma i*}(r_{1})f_{j}^{\sigma j*}(r_{2})f_{j}^{\sigma j}(r_{1})f_{i}^{\sigma i}(r_{2})\delta_{\sigma_{i}\sigma_{j}}\right]$$

The C.E non-interacting expression is thus:(22)$$U_{S}^{QM}=\frac{1}{2}\iint \frac{n\left(r_{1}\right)n(r_{2})}{\left|r_{1}-r_{2}\right|}dr_{1}dr_{2}+\frac{1}{2}\iint \frac{J_{s}\left(r_{1},r_{2}\right)}{\left|r_{1}-r_{2}\right|}dr_{1}dr_{2}$$

The C.E non-interacting in the Kohn–Sham scheme, by creating free self-interaction, can be attained by substituting the traditional Hartree energy (Equation (4)) by quantum mechanical expression (Equation (22)); the energy functional (Equation (1)) is as follows:(23)$$E\left[n\right]=\int v\left(r\right)n\left(r\right)dr+T_{s}\left[n\right]+U_{S}^{QM}\left[n\right]+E_{c}\left[n\right]$$

This $E_{xc}\left[n\right]$ at that point represents the correlation energy only, which is the variance amongst the Kohn–Sham and the exact system, or is seen as the variance between adopting the non-interacting and the interacting density pair. Correspondingly, as designated above, the expression for the Kohn–Sham potential is thus:(24)$$v_{KS}\left(r\right)=v\left(r\right)+\frac{\delta U_{S}^{QM}\left[n\right]}{\delta n\left(r\right)}+\frac{\delta E_{c}\left[n\right]}{\delta n\left(r\right)}$$

The responsibility is afterward to calculate the functional derivatives in regard to the density. The correlation functional form is largely unfamiliar, such that rough calculations are performed. Hence, we focus on the resolution/estimation of the functional derivative for non-interacting C.E. Considering the Hartree scheme, the derivative denotes an insignificant challenge since the equation hinges obviously upon its density as well as one acquires the renowned Hartree potential thus:(25)$$V_{H}\left(r\right)=\frac{\delta }{\delta n\left(r\right)}\frac{1}{2}\int \frac{n\left(r_{1}\right)n\left(r_{2}\right)}{\left|r_{1}-r_{2}\right|}dr_{1}dr_{2}=\int \frac{n\left(r_{1}\right)}{\left|r-r_{1}\right|}dr_{1}$$

Conversely, the exchange term functional derivative (Equation (11)) is not apparent for the reason that its representation hinges only upon the density implicitly via the reliance of the orbital’s density. One alternative approach adopted thus far in obtaining the derivative in question is the OEP (optimized effective potential) [25,26].

Of late, a group of researchers suggested a substitute and facile approach for obtaining the exchange potential [27,28]. The elementary impression is enlarging the employed Kohn–Sham orbitals, *f _j_^σ^*(*r*), that form the density, in its whole along with its orthonormal base set [29,30], also referred to as equidensity basis or Harriman orbitals, whose elements are inscribed explicitly with respect to the density permitting the presentation of functional derivatives relating to the quantity. The authors, as in their recommended approach, expanded the Kohn–Sham orbitals on the aforementioned foundation:(26)$$f_{j}^{\sigma }=\underset{K}{\sum }a_{k}^{j,\sigma }\emptyset_{K}^{\sigma }\left(r,\left[n^{\sigma }\right]\right)$$

Here the *a*_*k*_*^j^*^,*σ*^ represents expansion coefficients along with the equidensity basis elements, $\emptyset_{k}^{\sigma }\left(r,\left[n^{\sigma }\right]\right)$, presented as:(27)$$\emptyset_{k}^{\sigma }\left(r,\left[n^{\sigma }\right]\right)=\sqrt{\frac{n^{\sigma }\left(r\right)}{N^{\sigma }}}e^{ik.R^{\sigma }(r;\left[n^{\sigma }\right]}$$

The above expression is signified with a collection for endorsed integers *k* = {*k*_1_, *k*_2_, *k*_3_}. *R* is well-denoted by standardized integrals across its density, consecutively integrating across its coordinates. For specifics in the Cartesian coordinates, readers are referred to earlier work [27].

Aimed at calculating the exchange potential, the authors obtain the derivative of *J_s_* (*r*_1_, *r*_2_) in regard to the density by substituting the orbitals within the exchange term with its extended forms (Equation (16)) along with using the product rule in obtaining the derivatives.

## 3. DFT Concept in Graphene/Graphene-Based (Polymer)Composite Materials (GPM or GPCM)

Graphene and its materials rGO, or GO (graphene oxide (GO) as well as reduced graphene oxide (rGO)) are 2D (two-dimensional) slabs/sheets of sp^2^ and/or sp^3^ carbons organized in six-grouped rings that could be chemically altered to encompass an assortment of spontaneously active functional spots [31,32,33,34,35,36]. The utmost utilized forms of graphene materials are the single, bi-layer/multi-layer graphene, including rGO, or GO. Single-layer graphene can be prepared via repeated mechanical exfoliation [37], meticulous growth on a substrate such as SiC [38] through CVD (chemical vapor deposition) [39]. GO is the oxidized form of graphene, chemically altered by carboxylic acid, hydroxyl, and epoxide groups on the slab/plane [37,38]. The carboxylate functional entities offer the pH-dependent surface charge and colloidal stability [40], whereas the -O- (epoxide) and -OH (hydroxyl) groups can interrelate through H-bond (hydrogen bonding) [40]. Graphene oxide (GO) is an amphiphilic molecule that can be utilized as a surfactant in stabilizing hydrophobic molecules in solutions [41]. rGO can be obtained by ultraviolet (UV) and thermal treatment of GO in a reducing condition [39]. rGO is largely synthesized with the aim of restoring the electrical character as well as the optical absorbance of GO as the surface charge, oxygen functionalities, along with hydrophilicity are reduced [37,38,39,40,41].

The DFT concept in graphene/graphene-based (polymer)composite materials (GPM or GPCM) has been effectively utilized by several researchers globally [42,43,44,45,46,47,48,49,50,51,52]. In one of the instances, Salmankhani et al. [32] presented a classical simulation algorithm for graphene-based materials in their report. At first, *H_2_S* molecules’ structural adsorption upon BeO, ZnO, graphene, as well as Ni-ornamented-graphene were simulated along with geometrical optimization using DFT computations by means of SIESTA (Spanish Initiative for Electronic Simulations with Thousands of Atoms) computer coding [31]. For the purpose of studying the electronic performance as well as associated effects, the authors embraced the GGA. They pointed out that conventional DFT methods are unsuccessful in exactly addressing the van der Waals (vdW) forces without revealing dispersion interactions, in spite of their decisive character in feebly interfacing systems [31,32]. Hence, the basic DFT for GGA calculation was thus supplemented with the ab initio vdW tactics obtainable with Grimme, denoted as the vdW-DFT technique. This permits the amalgamation of dispersive vdW interfaces into DFT. A group of authors used the split-valence binary-ξ BS of the localized numerical atomic orbitals through the modeling procedure as well as polarization functions (DZPs) [42] where the energy-shift was set as 50 meV and the split-norm to 0.25: Additionally, the authors used a 5 × 5 × 1 Monkhorst-Pack grid for the k-points sample selection of BZ, whereas the atomic positions were relaxed while waiting for the atomic entities outstanding forces to be less than 0.02 eV/Å with respect to previous work [43]. Furthermore, to examine the CD, the mesh cut-off was set as 120 Ry by adopting BS Superposition Error (BSSE) tweaks through the addition of phantom atomic entities for the entire stages, which allowed the calculation of the isolated adsorbent on graphene [42,43,44]. The practice of BSSE corrections is essential for such categories of calculations aimed at providing precise energies when the system atoms interact [44]. In another instance, a periodic boundary state has been imposed on graphene and graphene-like surfaces in a way that the vacuum height was selected to be equal to 20 Å [45]; the value that could abolish slab–slab interactions where the selected supercell was composed of 50 atoms [42,45]. The structures consisting of adsorbents plus adsorbates have been shown to be at first simulated in diverse feasible structures and then permitted to ease thru the entire optimization process [42]. As per BSSE tweaks, the *H_2_S* molecule adsorption energies interface with BeO, ZnO, pristine graphene, as well as Ni-decorated-graphene surfaces is determined with the use of the following expression [42,43,44]:(28)$$E_{int}^{BSSE}=E\left(\frac{Sur\;f}{H_{2}S}\right)-E\left(\frac{Sur\;f_{ghost}}{H_{2}S}\right)-E\left(\frac{Sur\;f}{H_{2}S_{ghost}}\right)$$
where $E_{int}^{BSSE}$ represents the interaction energy and $E\left(\frac{Sur\;f}{H_{2}S}\right)$ denotes the nanosheet surface (*surf*) entire energy intermingling with the *H_2_S*. The “ghost” in $E\left(\frac{Sur\;f_{ghost}}{H_{2}S}\right)$ and $E\left(\frac{Sur\;f}{H_{2}S_{ghost}}\right)$ terms represent the counterpoise alterations utilizing “ghost” atomic particles. In particular, the terms agreed with the supplementary basis wave functions cantered at the *H_2_S* molecule position or the slate (nano-sheets’) surface, nevertheless deprived of the atomic potential. The authors deduced that the negative value of $E_{int}^{BSSE}$ echoes an energetically steady adsorption arrangement. All the schemes in their work were carried out at a temperature (constant) of zero Kelvin, although in the investigation the essential temperature should be counted as an active factor, making it unreasonable in comparing their theoretical findings with the experimental results, and so the concluding result was not provided in their report [42].

In another study, DFT analysis of NiO@Graphene nanocomposite as a catalyst for oxidation of urea in an alkaline electrolyte by Lu et al. [34]: this composite could be an excellent candidate for EMI shielding. Mashhadzadeh et al. [35] using DFT have shown the investigational as well as quantum mechanics multiscale simulation of mechanical performance of a PVC-graphene amalgamated system. These authors in their theoretical archetypal presented a multifaceted model for finite element utilized for the prediction of the composite’s Young’s modulus: they also modeled pristine graphene’s molecular structure using a DFT technique where they assumed graphene to be the interplanetary-frame architecture preserving the discretized graphene nature and then simulated by means of 3D flexible beam elements for carbon–carbon covalently attached bonds as well as atoms centered point mass elements. At the point where they claimed that interfacially interacting vdW interfaces occurring among the polymer-matrix (PVC) along with graphene were schemed by means of general form Lennard–Jones potential as modeled with non-linear truss rod standard. Lennard–Jones factors and van der Waals forces in their work were justified against the partition void for the cited non-linear truss rod via the DFT technique. Lastly, they fabricated PVC-graphene nanocomposite samples with diverse wt.% of graphene nanoplatelets empirically utilizing the melt-mixing approach. Their report revealed that the computational modeling established that the magnitudes of Young’s modulus PVC/graphene closely agreed with the experimentally obtained results until 1 wt.% showing an average difference of ~25%. Finally, they validated the mechanical findings obtained through the investigation of the morphology of the prepared experimental specimens by means of transmission electron microscopy (TEM) as well as scanning electron microscopy (SEM) micrographs [35]. The prediction of functional, mechanical as well as the structural character of graphene-polymer composites has been a point of attraction of study for several researchers globally within the last decade [31,32,33,34,35,36].

We project that the graphite-C3N4 family is an important group of materials in the present dispensation; understanding their simulation with the aid of DFT towards EMI shields fabrication and application is vital. A number of works in the literature have reported DFT utilization, although not for EMI shields in these materials systems [53,54]. In this light, g-C_3_N_4_ nanoribbons characteristics, as well as the correlation between the structural architecture or other factors and their properties, have been investigated by means of DFT [55,56,57,58]. 

Information about DFT calculations for other carbons towards property enhancement has been reported in a number of works in the literature within the last decade [59,60,61]. Oligothiophene dyes (OT) deposition on diverse carbon hybrid materials have been theoretically and experimentally investigated with the aid of potent DFT technique where these authors explained the interfacial interaction at a molecular level for thiophene; 2,2′:5′,2′′-terthiophene as well as α-sexithiophene with CNTs and established their studies with theoretical DFT modeling/calculations using diverse tubules lengths, sizes, along with hydrogenated open ends [60]. These researchers used OT entities as well as CNT geometry optimization by means of the B97-D/6-31G(d,p) technique. The thiophene oligomers containing CNTs DFT calculations comprised of about 6564 basis functions. B97-D; B97-D3 and wB97x-D coupled 6-31G(d,p) density functionals basis set were utilized for investigating the interfaces between (7,7) SWCNT and OT dyes. Their DFT investigation involved the armchair nanotubes configured on either end with saturated hydrogen atoms [60], while the interactive energy estimation was performed using the supermolecular method by means of counterpoise correction contained within. The entire quantum chemical calculations were carried out by means of the Gaussian 09 software package. Seeing that more than a few carbon materials may be modeled in a similar approach, graphene or graphene-based composites can be modeled similarly.

One other important point with the use of DFT in graphene systems is that DFT calculations can be utilized to study the influence of graphene size/length on its properties [60,61]. In this regard, Tyagi et al. investigated twin dissimilar models of ovalene (C_32_H_14_) as well as circumcoronene (C_54_H_18_) along with their corresponding doped-models (C_31_XH_14_, C_53_XH_18_ given that X = Pt, B, Ni, Fe, P, N, and Al) at GGA-PBE/DNP. In comparison to diverse structural estimated parameters as well as electronic properties, these models were studied with respect to the electronic density of states (DOS) spectra plotted in order to ascertain visible changes in the electronic characteristics as the size increases. These authors reported that there were no significant variations in structural and electronic character with progression from the smaller model to the higher one. It is found that doping maintains the planarity of the surface but induces comparatively large changes in bond lengths about the doped-atom, thereby resulting in bonds weakening [61].

### 3.1. (Electronic)Structures and Atom Projected Density State of Graphene and/or GPM

The understanding of the electronic structures and atom projected density state of graphene and/or GPM is very important in determining the application area among different researchers and/or industrialists globally. Graphene is well known as a honeycomb-lattice structured solitary carbon atoms layer having two equivalent atoms crystallographically “C1 and C2” in its elementary unit/elementary cell. Its *sp*^2^ hybridization amid one 2*s* orbital along with two 2*p* orbitals results in a trigonal planar architecture, including the creation of tough bonds between the atomic carbon entities that are 1.42 Å apart. The bands possess a fully occupied shell resulting in a profound valence band. The unaltered 2*p_z_* orbital, that is at right angles to the planar assembly of the graphene layers, could covalently bind with adjacent C-atoms, which leads to the creation of π bands. Every single 2*p_z_* orbital has an additional electron, meaning the π band is half filled. The π and π* bands touch themselves in a solitary position at the Fermi energy (*E_F_*) towards the extreme-spot edge of hexagonal graphene’s BZ, as well as nearby to the purported DP; the bands show a direct dispersion and result in perfect Dirac cones. Hence, undoped-graphene is a semi-metal also known as a “zero-gap SCD”. These bands’ linear dispersion results in a quasi-particulate matter having zero mass, also referred to as Dirac fermions.

However, this unique graphene “0 gap” electronic structural architecture results in a few limitations in application in real-life electronic devices. For instance, with a focus on the preparation of real-life transistor, graphene layer having induced energy band gap through using electric field (EF) or by the adjustment of its electronic structural architecture using diverse functionalization approaches including the adoption of diverse substrates that alters graphene’s electronic structure; amalgamation within the graphene structure nitrogen, boron as well as transition-metal atoms; intercalation of predetermined substances beneath graphene grown on diverse substrata/fabrics; deposition of molecules/particulate matter on graphene top, and so on.

Authors [47] studied the effect of a sequence or groups of graphene-functionalized additives for insulation of power cables to suppress the build-up of an electrical pre-breakdown phenomenon in solid insulation (electrical treeing) and thwarting the polymer matrix degradation using DFT theory. They investigated the Bader charge which revealed that pristine, doped-graphene showed successful detention of blistering electrons and restrained their assault on the crosslinked polyethylene (XLPE) due to π–π conjugated-unsaturated formations. Supplementary investigation of the materials electronic character within its interfacing area intermediate the active ingredients and XLPE displayed that the N-doped single-vacancy active ingredients (graphene, graphene oxide and B-, N-, Si-/P-doped GO) presented comparatively a robust physiochemical interface with the XLPE in restricting its movement and somewhat feeble chemical activity to restrain C–H/C–C bond cleavage, which suggests they were all potential active ingredients [47]. They said the comprehension of functional group-enabled graphene ingredients features actively in arresting electrons and the interfaced contacts capable of assisting in the auspicious additives screening as power cables voltage stabilizers. The authors use a (6×6) supercell to construct the model for the virgin graphene platelets (G), where the models of GO, GO-doped N, Si, B, as well as P atoms (Si-GO, N-GO, B-GO, and/or P-GO), along with SVG (solo-vacancy graphene) and SVG doped N or B atoms (N-SVG, B-SVG) were constructed and optimized, as depicted in Figure 1 [45]. They adopted a chain-like 4-methylheptane, C_8_H_18_ (CH_3_CH_2_CH_2_CH(CH_3_)CH_2_CH_2_CH_3_) organic compound to model the chemical action of the polyethylene molecule on the series of graphene sheet surfaces. Their model was mainly focused on the simulation of the C–H bond-breaking behavior of C_8_H_18_ by the catalytic action of a series of graphene sheets along with the local interface among them, which could successfully reveal the core interface amongst the XLPE and graphene sheet reinforcing-fillers within the real-physical challenge [47]. In another study, Gorb and co-workers [49], adopted an augmented density functional theory through long-range adjusted hybrid density functional 6-31G(d,p) and a ωB97XD basis set which was applied to create sandwich structures consisting of nanocomposites between graphene oxide and a polymer matrix (polyvinyl alcohol(PVA)). These authors predicted the interface energies and conversed the involvement of electrostatic and dispersion constituents in the final composite structural and functional properties [49]; along with the computational generation of IR spectra of intercalates and their comparison with those obtained experimentally. They suggested two sources of interfacial energy for stabilizing the intercalates between GO and PVA as the electrostatic and dispersion (van der Waals) constituents [49]. They predicted structural properties of the synthesized material for IR, XRD, etc. successfully [49].

Özkaya, and Blaisten-Barojas [50] established for their study of polypyrrole on graphene using DFT that the results emphasize that polypyrrole (PPy) physisorbs onto the graphene surface; PPy physisorption on graphene happens at a distance of 3.5° A and a 23° tilt angle “*α*” (Figure 2); the PPy-graphene nanocomposite band structure does not possess an energy band gap opening at the Dirac point; PPy-graphene composite physisorption produces an electronic density rearrangement along the polymer backbone only; and a second graphene layer atop of PPy does not disturb the aforementioned results [50]. 

Reports have shown that there exist more than a few possibilities on how polypyrrole (PPy) adsorbs onto graphene surfaces; like physisorption of the polymeric matrix without altering the intrinsic properties of both materials. Another possibility is the chemisorption of the polymer matrix (PPy) created via the innovative bonds with graphene-based C-atoms. While the 3rd involves a reaction where new products are formed. The physisorption of PPy onto graphene surfaces normally happens at room as well as lower temperatures, although the remaining two scenarios involve elevated temperatures (or more external influences). Consequently, on average, the thermodynamics unveiling the correlation amongst graphene electronic structure with PPy physisorption states provides means for enhancing the stability of the composite. Özkaya and Blaisten-Barojas established a thorough first-principles technique for defining the energetic stability, electronic structure, as well as the molecular geometry of PPy, adhered onto graphene [50]. They predicted the physisorption of the polymer chains at 60° orientation in regard to the graphene hexagons in the bridge, top, or hollow sites with −0.25 eV/monomer binding energies at 3.5 Å distances. The PPy chains plane was reportedly skewed by ~23° with regard to graphene’s surfaces in the entire three sites. The graphene band structure was not significantly affected by the polymer layer and void of opening new band-gap energy at the Dirac point. Thereby, confirming that the physisorption of PPy chains between two graphene sheets did not affect their earlier reports [50]. They also observed that the absorption sites’ redistribution of CD occurring indicated a slight redistribution of CD as well as the PPy *π* orbitals leaving the graphenes’ surface electronic density unaltered. With consideration of this novel report, it can be agreed with the authors [50], that PPy/graphene composites/hybrids create new possibilities for designing advanced systems having high-tech interest along with the exploration of essential bottom-up synthetic chemistry towards applications in sensors, EMI shields, energy storage, conducting composites, fluid separation, automotive, electronic, along with biomedical industries.

### 3.2. Band Gap, Band Structures and Atom Projected Density States of GPM

Different materials are abundant and occur in diverse forms on Earth as an integrated facet of the present-day industrialized society. Among these materials, graphene/carbon-derived systems are very much essential for survival for plants, animals, and human beings. A plane monolayer C-atom tightly crammed into a 2D honeycomb-lattice assembly referred to as “graphene”; a material that acts as the foundational structural block of graphitic materials [51,52,62,63,64,65,66,67,68,69].

Adjoining carbon iotas sp^2^ crossbreed/hybrid within every C-molecule resulting in ternary covalent bonds emerging from the orbitals blend of its s, px, as well as py. Such adaptation saves three free electrons. The pz orbital holds this free electron; also this p-orbital rests over the plane and structures the π bond. This pz orbital assumes a huge part in the physical and synthetic properties of graphene. The formation process of graphene can result in different dimensions such as rolling, wrapping up, as well as stacking into 3D graphite, 2D GO/rGO, 1D nanotubes, as well as 0D fullerenes, respectively. This has attracted specialist attention because of its uses in different niches like biomedicals, composite materials, sensors, microelectronics, and so forth; from the time when it originated in the early 2000s [70] through an exfoliation strategy. The exceptional physical, electrical, and optical properties of graphene propose it as a planned contender for use in SCD electronic gadgets: it displays likewise excellent charge transporter portability (106 cm^2^Vs^−1^) which further grows its degree for use in SCD electronic gadgets [70]. 

Doping as well as adsorption of appropriate contaminant molecules regularly adds to the arrangement of a band gap in gapless graphene. Substitutional doping of heteroatoms or atoms can be alluded to as substance doping bringing about changes in graphene cross-section structures [70]. The points achieved by these adjustments are profoundly reliant upon the sort of dopants, their focuses, and positions inside the graphene framework. The graphene-doped structure Fermi level might go down or up from a zero changed level contingent upon the electronic nature of the dopant(s). The electron inadequate unfamiliar doped molecules comparative with that of atoms of C affected by Fermi level to shift descending accordingly showing p-type doping. Then again, electronically rich subbed particles improve the Fermi level appearance n-type conduct [70]. 

The band gap tuning of graphene can be effectively achieved by substitutional doping of components like Be-O, Be, B, N, Al, Mg, Br, and O-molecules. The impact of dopant fixation subbed positions, and their effect on mathematical boundaries have been discussed in detail [70]. The adsorption of unfamiliar particles frequently permits graphene’s electronic qualities to change, where the recommended approach is profoundly adaptable because of band holes, and the type of graphene groups can be tuned by changing the adsorption calculation, the inclusion of dynamic adsorbate molecule(s), or by synthetic adjustments of the adsorbate(s) [71]. Kozlov et al.’s DFT calculations revealed that the physisorption of particles with a specific electronic setup, recognized by the most reduced vigorously situated vacant atomic orbital nearby the Dirac point, opens the band gap and makes it conceivable to design band structures [71]. The underlying as well as electronic character of oxygen-adsorbed-graphene sheets utilizing first-standards absolute energy electronic construction computations inside the neighborhood DFT were examined by Natori and collaborators [72]. They reported that a limited energy hole arises for the oxygen-adsorbed-graphene and its increments with the proportion of O/C, as shown by tests too [72]. The impact of adsorption of pre-separated O-molecules just as substitutional doping on alteration in-band hole in graphene was examined by Hussain and Basit [70]. They recommended that the troublesome vigorous cycle of making opening ought to be smoothed out to achieve a similar objective of band gap tuning in graphene, where the adsorption system of sub-atomic oxygen (O_2_) on the ideal and faulty surface of the graphene is investigated [70]. These authors used first-principles calculations with the adoption of the Vienna ab initio simulation package (VASP) based on DFT and described the electron–ion interaction using the projected augmented wave (PAW) technique and the Perdew, Burke, and Ernzerhof (PBE) version of the GGA adopted for the exchange and correlation parts of the electron (e^−^)–electron interactions with the plane-wave cutoff energy set as 450 eV. They selected a 4 × 4 graphene supercell (32 atoms) slab model with the application of the periodic boundary conditions in all directions [70]. In studying the electronic structure of graphene using DFT with the avoidance of artificial interlayer interactions, the vertical separation between the two graphene sheets should be fixed at a value ≥14 Å. A Monkhorst–Pack grid may be adopted to sample the BZ with 5 × 5 × 1 [70], for the DOS denser K-point grids of 16 × 16 × 1. A first-order Methfessel–Paxton smearing function having a width of ≤0.1 eV could be adopted to account for the occupancies fractionally [70,73]. Energy minimizations of the graphene-based structures in DFT can be carried-out until the whole Hellmann–Feynman forces are <0.01 eV/Å, with the relaxation of all the atoms in the systems throughout the process geometry optimization [70]. In order to give a numerical account of the doped system’s strength, cohesive energy can be estimated by use of the expression:(29)$$E_{coh}=\frac{\left[E_{tot}-n_{i}E_{i}\right]}{n}\;\;\;\;\;\;(i=C,O)$$
where *E_coh_* represents the cohesive energy per atom of virgin and/or O doped conformations. *E_tot_* as well as *E_i_* denotes the entire energies of the assembly and/or the discrete elements available within similar supercell correspondingly. Lastly, *n**_i_* is the number of *i*th species available in the conformation, while the total number of atoms is *n*. Spin-polarized computations are then executed to ascertain the magnetic challenges. Hussain and Basit [70], found that spin-polarized and non-spin polarized computations presented similar findings as graphene is a non-magnetic material. Hence, computations carried out in this vein have been said to be limited to non-spin polarization for simplicity of computation [70].

With regards to graphene-based composites, the graphene-polypyrrole nanocomposite-film band structure has been studied as reported by Özkaya, and Blaisten-Barojas [50] using DFT calculations. They observed that the fundamental phenomenon of the PPy adsorption further investigated using computations of the band structure as well as DOS as shown in Figure 3 is the band structure of pristine graphene and PPy/graphene polymer composite estimated within the PZ-GGA technique. Their calculations revealed that virgin graphene exhibited zero band gap, as expected [50]. Their overall band structure for graphene (3 × 3), as they reported, was in agreement with other theoretical report findings [52]. They denoted that band gap absence at the Fermi energy was initiated by the merging of valence and conduction bands at the Γ-point in the Brillouin region/zone, and the typical linear band and a secondary band are exhibited by two graphene bands. These authors also reported that the band structure around the Dirac point adsorption was affected by the PPy chain via opening a very slight energy gap of 0.008 eV on the secondary band, although not on the graphene proto-classical linear band [52]. Nevertheless, as depicted in Figure 3, the DOS of graphene and PPy on graphene are different, most particularly by the emergence of the peak below the Fermi energy due to the PPy presence; the anticipated density of states (PDOS) upon the PPy atoms indicates this peak is due to the p-orbitals of PPy carbon atoms as in Figure 3b, revealing the weak *π* orbital interface with graphene. Furthermore, the authors stated that the PDOS demonstrated that the p-orbitals of PPy carbon and nitrogen atoms influenced the bands between −2 and −3 eV, whilst their s-orbitals give rise to the DOS structure revealed between 2 and 4 eV [52].

It was revealed that overall, the band structural architecture of PPy/graphene nanocomposite film presents the composite as a semi-metal, hence retaining the semimetal properties of graphene even though pristine PPy is well known to be an insulator having a band gap of 3.17 eV [52]. It is proposed that the semi-metal character of a nanocomposite like PPy/graphene composite does not rely on the DFT type utilized if such technique replicates the pure graphene band structure [52] properly. Investigations were also carried out on the effect of the band structure as a result of a second graphene layer laid atop the previous adsorbed PPy chain [52], where three feasible separations (8 Å, 11 Å, and 14 Å) between the graphene layers were taken into consideration: also, a vacuum zone above the restricted PPy chain was set at 17 Å. It has been postulated that a graphene bistratum arrangement with double inequivalent graphene stratums was revealed to present a finite energy gap at the Dirac point (DP) [62]. Additionally, the band structure of PPy/graphene within the proximity of the Fermi energy is substantially unaffected by the second graphene layer and the total DOS presented no significant changes in connection with the case of a single graphene stratum near the Fermi energy [52].

Periodic boundary conditions (PBC) modeling in polymer composites is of great importance, especially for graphene–polymer composite systems aimed at innovative applications, especially EMI shielding in our case. For graphene systems, we recommend the boundary conditions as per available literature, 3D periodic boundary conditions could be used for two-unit cells along with a four-atom-cluster as well as a 32-atom-cluster [73,74,75,76]. However, reports partnering to graphene-polymer a composite system have reportedly utilized 32, 162, etc. C-atom supercells for graphene along with a vacuum value ranging from 20 Å, 30Å, etc. aimed at omitting the interfaces between the adjacent supercell images [73,74,75,76]. The periodic boundary condition may be chosen as per the property aim at by the researchers in the DFT study: diverse graphene supercells comprising of C-atoms for PBC have been reported [73,74,75,76]. In a recent study, Meon et al. investigated the influence of the adopted modeling approach upon the boundary condition as well as the impact energy of low-velocity impact on CF reinforced polymer composite laminate [77]. The authors established that the PBC adopted in the fabric plane modeling presume a strong enough as well as perfect interface between the carbon atom at the edge with polypropylene chains in graphene–polymer composite systems [75]. These authors also represented the adsorption energy of the reinforcing graphene unto propylene as per the modeling as [75]:(30)$$E_{Ad}=E_{PP+G/TSWG}-E_{PP}-E_{G/TSWG}$$

$E_{PP+G/TSWG}$ stands for the structure total energy, comprising of graphene as well as propylene, while $E_{PP}$ and $E_{G/TSWG}$, correspondingly, represent the free-standing propylene-graphene overall energy. The gap between graphene and propylene was varied for the purposed of establishing the adsorption distance vs adsorption energy correlation, towards calculating the adsorption energy systematically. The nearest expanse between the graphenes’ carbon atom, as well as the hydrogen atom of the methyl group in propylene molecules, was defined by graphene sheet to propylene separating distances [75].

Equation (30) could be represented for the interfacial interaction between graphene and other polymeric materials in DFT calculations by substituting the concerned polymer for PP.

### 3.3. Effects of the Electric Field upon the Properties of Graphene-Based Composites: EMI Shielding Focus

The EF created within and around the graphene-based composites of any form affects its electrical and/or magnetic properties and, ultimately, the EMI shielding properties of the material. 

The influence of the EF on the resultant properties of ZnO@graphene nanocomposites with the help of DFT has been reported by Geng et al. [53]. Aimed at validating the existence of inter-material interacting charge transfer between Ag_2_CrO_4_, GO, as well as g-C3N4, 3D charge density difference graphical plotted for the interfaced materials entities via CGA heterostructure was utilized [78]. These authors detected that the e^−^(s) are firstly amassed within the g-C_3_N_4_, whereas they are depleted in the Ag_2_CrO_4_, signifying that the interface developed an IEF (internal electric field). At that point, the e^−^(s) amassed in g-C_3_N_4_ drifted to the GO, further extricating e^−^-hole pairs. Also, the planar average CD difference conformed to the CGA heterojunction, and substantial deviations in the CD variance at the composite interface were observed due to the IEF between Ag_2_CrO_4_, GO, and g-C_3_N_4_ [78]. These researchers also reported that the value of Δ*ρ* was an indication of e^−^ build-up or depletion, as the direction of IEF could be identified by calculating the difference observed. They also established that the e^−^(s) were inclined to transcend from Ag_2_CrO_4_ to g-C_3_N_4_ about the CA interface, even though they were apt to move from g-C_3_N_4_ to GO at the interface between CG [78]. Based on their conclusion that this kind of constant secondary e^−^ transmission mechanism base Z-scheme heterostructure will significantly enhance e^−^-hole pairs separation ability, resulting in heightening of the PEC property performance, it is envisaged that a similar phenomenon will occur in other GO–metal oxide composites. If this phenomenon can be substantiated, then such composites are envisaged to be potential candidates for EMI shielding and other advanced applications. In combination with DFT results, the probable charge transfer course of the CGA altered electrode using irradiation was established [78]. The resultant IEF energies photogenerates e^−^(s) within the conduction band (CB) of Ag_2_CrO_4_ flow to the valence band (VB) of g-C_3_N_4_. Furthermore, they established in their study that, the −0.49 eV e^−^(s) potential of GO is less than that of g-C_3_N_4_; hence it is thermodynamically useful for photogenerated e^−^(s) transfer to GO, from the CB of g-C_3_N_4_ [78].

## 4. Application of DFT in GPM as per Literature 

### 4.1. General Application to/in Atoms

It is important to highlight that an exchange can be treated precisely when surrounded by the optimized potential technique. With regards to atoms and atomic ions, the DFT technique could be adopted to enhance the mathematically rather elaborate approach as estimated by Krieger et al. [79]. Table 1 provides exchange energies for diverse closed-shell-atoms. The gradient exchange functional gives considerable enhancement across the LDA (local density approximation). Table 2 shows the summation of energies [80]. As is normal, the Hartree–Fock total energy is to some degree lower than that acquired from the exchange-only OPM 48, 131; however, the difference is negligible. It is observed over time that the LDA does not perform poorly as expected: its error is in a similar order as the error observed in the Hartree–Fock, although in a direction opposite to it. With regards to GGA, considering the findings above the experimentally obtained results, the precision is enhanced significantly.

Looking at the correlation, with respect to Gross and co-workers [83], the correlation energies of orthodox density functional as well as quantum chemical function can be evaluated. The correlation energy of any compound in quantum chemistry is conventionally referred to as the variance between the correct (non-relativistic) energy and the entire Hartree–Fock energy:(31)$$E_{C}^{DC}=E_{exact}-E_{HF}$$

Conversely, in the density functional concept, the correlated energy is gotten by injecting the precise base-state density into the correlation functional ($E_{c}^{DFT}$):(32)$$E_{c}^{DFT}=E_{c}\left[n\right]$$

This is presented by the difference of the precise exchange-correlation along with the exchange energies:(33)$$E_{c}^{DFT}=E_{xc}^{DFT}\left[n\right]-E_{x}^{DFT}\left[n\right]$$

We should remember that the exchange energy derived from the density functional may be represented using the Hartree–Fock model designed for exchange energy which reveals that the precise injection of the Kohn–Sham orbitals:(34)$$E_{x}^{DFT}=E_{x}^{HF}\left[n\right]-[U_{i}^{KS}]$$

The density functional correlation energy is presented as $E_{c}^{DFT}=E\left[n\right]-E^{HF}[U_{i}^{KS}]$, since *E*[*n*] = *E*_*exact*_, even though the quantum chemical correlation energy is represented as
(35)$$E_{c}^{QC}=E\left[n\right]-E^{HF}[U_{i}^{HF}]$$

Meanwhile, the Hartree–Fock orbitals are used to reduce the Hartree–Fock total energy, resulting in the inequality; $E_{c}^{DFT}\leq E_{c}^{QC}$. Nevertheless, the variance between the two correlation energies is negligible [84], as depicted in Table 3. 

Therefore, with the utilization of the DFT technique, the variation between the different energies within an atom or compound can be revealed.

In Table 4, the computed correlation energies by diverse functionals approximation are presented: the WL (Wilson–Levy) [85], LYP (Lee–Yang–Parr) (LYP), the GGA (Perdew and Wang) [86] the LW (local Wigner functionals) [82] along with the local correlation functional of Perdew and Wang (LDA) [87].

### 4.2. Molecules

The DFT has been established into a cost-effective wide-ranging technique of estimating the molecular character of molecules. The Kohn–Sham theory applied to chemistry as per the literature is vast, although it seems impractical to give an entire account of the review articles on molecular computation techniques. However, considering a few works of literature, correlation energies for 21 closed-shell molecular entities estimated via Wilson–Levy are presented in Table 5 [82], the Lee–Yang-Parr [89], the local Wigner, as well as the Perdew-Wang functionals. With regard to all these molecules, their functionals generate an acceptable assessment of the correlation energy generated experimentally [90]. It has been shown that substantial variations exist within the correlated energy density, particularly the Wilson–Levy scheme which results in a local performance relatively different from the rest [82].

With the help of DFT, it is revealed that the local-density approximation results in severe overbinding (consider the atomization energies in Table 6), although other quantities like the length of the bond, bond angles, and the vibrational frequencies have a tendency to quite well agree with the experiment [82]. Also, the bond length, for instance, some first-row diatomic molecules as represented in Table 7 is overestimated (mean deviation ~0.01 Å).

By use of the DFT technique, the correlation functional as per Lee–Yang–Parr [89], in combination with those of Becke [93], presents long bond lengths that are, to a certain degree, frequency vibrations which are often improved in comparison to MP2 as well as pretty acceptable energies due to atomization. By DFT simulation, it has been shown that simple molecules in the form of hydrides having lone-pair electrons (like NH_3_) have a tendency to fasten less, whereas manifoldly bonded molecules, like $F_{2}$ overly bind with this estimate.

It is known through DFT that the Becke–Perdew approximation is a development over the local density approximation, particularly in the case of single bonds [82]. However, for some XH-bonds, the length of the bond is extremely lengthy, and subsequently, the frequencies due to vibrational forces are negligible or too small. However, this functional(s) presents higher predictive precision for atomization energies.

Again, with DFT, the Becke–Roussel exchange functional in combination with the correlation functional of Perdew results in enhanced geometries for many molecules [82].

Within the last decade, a novel category of hybridized DFT (Hartree–Fock as well as density functional) techniques have been projected; its plainest form comprises of exchange-correlation energy, thus: (36)$$E_{xc}=E_{xc}^{DFT}+a\left(E_{x}^{exact}-E_{x}^{DFT}\right)$$

This parameter a is calculated by matching it with the thermochemically derived experimental data, as meticulously founded by Görling and Levy [94]. These authors utilized hybrid DFT techniques and revealed that the hybrid schemes rely on modelled schemes denoted as the Slater determinant that generates the precise ground state atomic density along with the minimization of the prospected operator value $\hat{T}+a{\hat{V}}_{ee}$ being a parameter possessing a value between 0 to 1. By use of this form of exchange intercourse, average bond energy errors are minimized significantly (towards ~2 kcal mol^−1^), and the barrier heights to the reaction are enhanced [82].

Enormous molecules like those of interest to biologists can be handled with DFT, although in the Hartree–Fock technique, the costs of computation rise as *N*^4^/*N*^3^, those in the Kohn–Sham model like *N*^3^. However, within DFT, in principle, it is possible to work out linearly scaled schemes for molecules and their composites [82]. The reason is that the entire electronic assembly is estimated exclusively with the aid of electron density, which has been a very vital tool/technique for molecular computations.

DFT has made it possible for large-scale mathematical computation of molecules and solids in graphene-based composite materials in EMI shielding applications and others.

DFT calculations have been utilized to investigate the interfacial architecture and properties of ZnO layers coated graphene subjected to the influence of outward EFs: the authors used three categories of structures, pristine graphene (P-G), graphene having a single-vacancy (V-G) as well as graphene with epoxy (O-G) to represent the surface morphology of the graphene having defects as well as GO [62]. The geometric structures and binding energy findings revealed that graphene interaction with ZnO occurred mostly via vdW interface, resulting in the development of a firm film architecture, even though these composites were reported to be stable under the influence of external EF [62]. 

The application of graphene-based composites/nanocomposites for diverse materials, especially, metal oxide@graphene, conducting polymer@graphene, etc. composites, and analysis of their potential in EMI shielding still has a great potential.

## 5. Challenges and Future Outlook

To date, the DFT technique has been posed/faced with several unresolved problems: Although DFT has turned out to be a widespread technique in several branches of material science, organic/inorganic chemistry, conventional DFT is not devoid of challenges [95,96,97]. Straightforwardness is the key characteristic of DFT, but the initiation of functionals as well as their approximations makes DFT problematic. The precise description of estimation and/or geometries of the binding energy of molecules is challenging with DFT. The challenge associated with the LDA/GGA category of functionals is their symmetrical underestimation of the state transition barriers of materials such as transition metal oxides. Furthermore, in most cases, London dispersion force and vdW free forces present a limitation for estimated functionals which makes precise and effective descriptions of London dispersion force and vdW free forces, covalent bonds, as well as state transition continuing challenges of DFT for such systems in an instance where the aforementioned interactives forces are actively acting concurrently. A self-interactive error occurs in DFT when deciphering a single-electron scheme because DFT takes into account overall density *ρ* without taking care of individualized electrons where a specific electron is considered to non-physically self-interact. Even though significant attempts have been made toward realizing improved performance upon numerous molecules/molecular sets, a significant extent of error is still linked with modern-day functionals in facile techniques. Functional collapse for infinitely overextended H_2_^+^ and infinitely overextended H_2_ within graphene-based structures is amongst the essential constraints in the contemporary electronic architectural scheme. Hence, constant improvements are compulsory to realize the appropriate explanation of the robust relationship, energy gap, and so forth.

Firstly, current theoretical simulations using DFT cannot effectively reveal the temperature change influence upon the properties of analyzed materials. For instance, DFT is not able to forecast the precise state of insulating for transition metal oxides [97]. Also, Hartree-Fock included functionals like hybridized functionals possess a higher inaccuracy compared to normal functionals as well as performing badly on stationary correlation challenges [97].

Secondly, up-to-date theoretical simulations are generally executed to make known the inherent character of a chosen material. The struggle has thus far been confined to evolving algorithms aimed at simulating realistic reaction settings, like surfaces that are charged along with solvent environments. Sooner or later, innovative algorithms should be established, making hypothetical simulations having greater accuracy; 

Thirdly, the DFT size for the simulation approach is restricted to nano-scale, making it hard for DFT to deliberate on the entire specification for a given experimental consideration. For instance, the simulation of a bulky metallic atom in a graphene-based (nano)composite is constantly parted into the selected base simulation as well as adjacent to reduce the computational problem. 

Principally, the Monte Carlo technique is founded upon the feasible accepted form of DFT computation and is expected to be an acceptable resolution towards acquiring mesoscale graphene-based (nano)composite simulations.

Fourthly, the fast advancement in computer studies may also encourage a revolutionary growth in theoretical modeling-based research. To accelerate the invention speed for new materials, greater interest ought to be paid to exploring universal descriptors for the residences of a number of materials, such as nanoscale and mesoscale graphene-based (nano)composites. 

Also, with the growing influence of “big data” and high-output computational calculations in materials, physical science, and chemical science, it is essential to assure high precision for accumulated first-principles data within data sets. Therefore, there is new interest in convergence facets of DFT estimations. There is abundant information on the convergence by means of the sampling of reciprocal space from prior work with significantly more restricted computational assets. For SCDs as well as insulators, this information about distinct *k*-points can be utilized to undertake the estimations in the most conservative manner. With regard to metals or their alloys, the precise test group of the Fermi surface is an issue and requires the utilization of compact k-point networks toward guaranteeing a union of the all-out energy per atomic particle to >1 meV. In any event, for straightforward metals or their alloys, there is a bewildering variety when looking at their Fermi surfaces. In this manner, there is as of now no generally acceptable formula accessible for picking *k*-points that bypasses a condensed Brillouin zone sampling. Besides, for high sampling bulk, the specific area of every k-point, and in this manner, the information about uncommon k-point sets, turns out to be less pertinent. On the off chance that the objective of the computations is the all-out energy, as well as the atomic structure (relaxed), the most effective method for managing the Fermi surface is a widening of the Fermi–Dirac appropriation function. The widening should be picked with the end goal that an adequately large part of all occupation numbers shows partial occupation. The strategies are accessible for extrapolating the absolute energy to the zero limit broadening expanding function, additionally with an exceptionally large number of *k*-points. The forces on the atomic particles merit unique consideration since they are determined as the subsidiary (derivative) of electronic free energy.

Again, studies involving magnetic properties of penta-hexa-graphene and their utilization in advanced materials, with proper simulations adopting DFT, are limited or rare [64,65]. This graphene structure is made-up of an amalgamation of hexagonal and pentagonal rings of carbon atoms, therefore, referred to as Penta-Hexa-graphene (ph-graphene). It possesses an antiferromagnetic (AFM) ground state which can be transformed into a ferromagnetic (FM) state under applied force/strain [95,96]. The latter state is shielded by a small strain-induced energy wall. These results could initiate advanced research to induce magnetism and spin-flip barriers through strain in other 2D metal-free materials. This is an area that ought to be investigated by researchers globally for the fabrication of state-of-the-art innovations.

## 6. Conclusions

With the adoption of powerful DFT techniques, achievements are reportedly being attained in the simulation and engineering of graphene-based (polymer)composites/hybrid systems along with other high-performance materials. The amalgamation of DFT computations with experimental findings is an encouraging research path in materials science. Currently, widely used density functional theory codes are ELK [77], quantum espresso [98], VASP [99], Siesta [100], Abinit [100], Wien2k [101], etc. Regarding graphene-based devices, this assessment has briefly covered DFT simulations in these materials. Also, a brief consideration of the structure as per DFT simulations and experimental results has been given. The application of DFT in graphene-based (polymer)composites/hybrids has also been discussed, along with the challenges and future scope. This article will shed light for researchers globally on the effectiveness of the use of DFT simulation in graphene-based (polymer)composites/hybrid materials for advanced applications. Plans geared towards the enhancement of this review in several chapters or phases are in place, given its uniqueness and importance to the research community dealing with graphene materials.

## Figures and Tables

**Figure 1 polymers-14-00704-f001:**
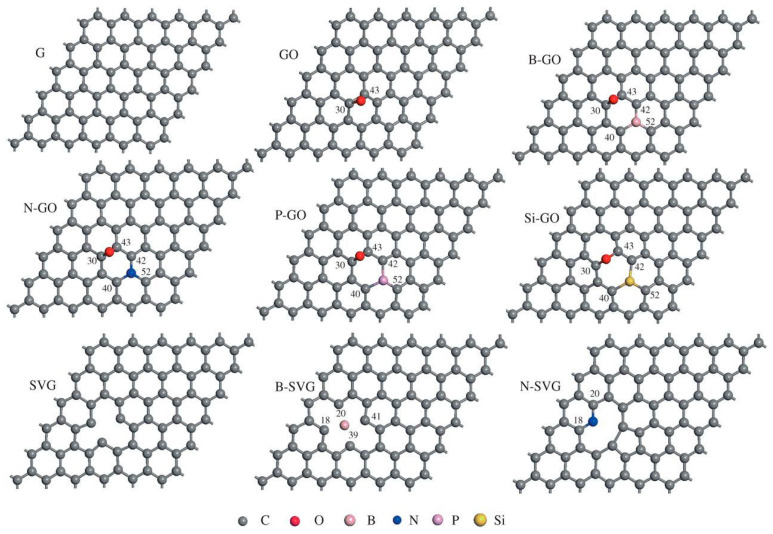
The supercell models of pristine and functionalized graphene sheets. Reproduced with permission from [47]. Copyright 2018, Royal Society Chemistry.

**Figure 2 polymers-14-00704-f002:**
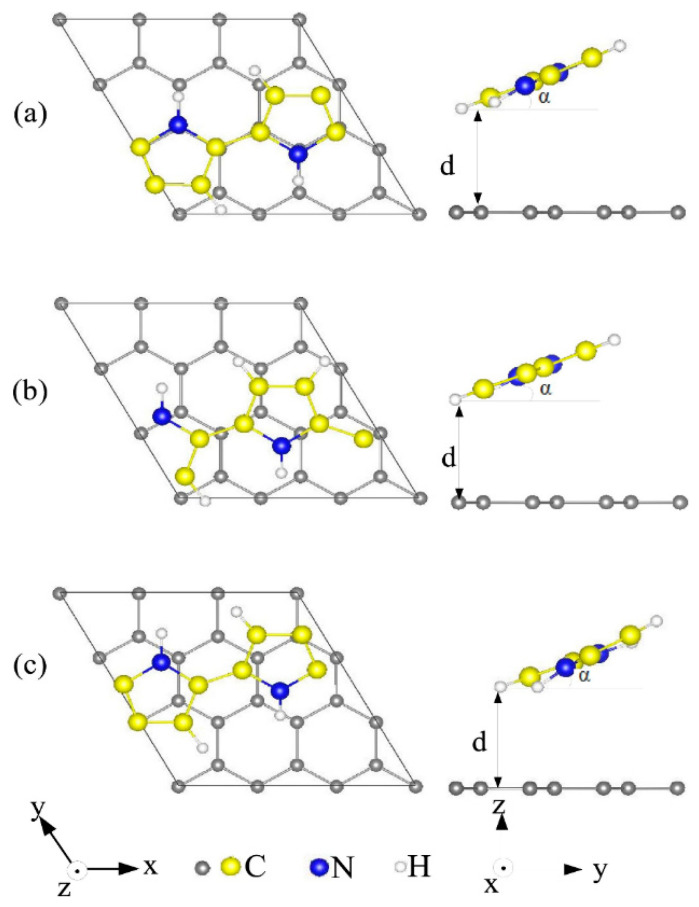
Top and side views of the optimized structure of the PPy/graphene sheets at (**a**) B bridge, (**b**) T top and (**c**) H hollow sites. Graphene carbon atoms are depicted grey. The PPy carbon atoms are depicted yellow, the nitrogen atom is blue, and hydrogen atoms are white. Side views show a distance d separating the PPy from the graphene surface. Reproduced with permission from [50]. Copyright 2018, Elsevier Science Ltd.

**Figure 3 polymers-14-00704-f003:**
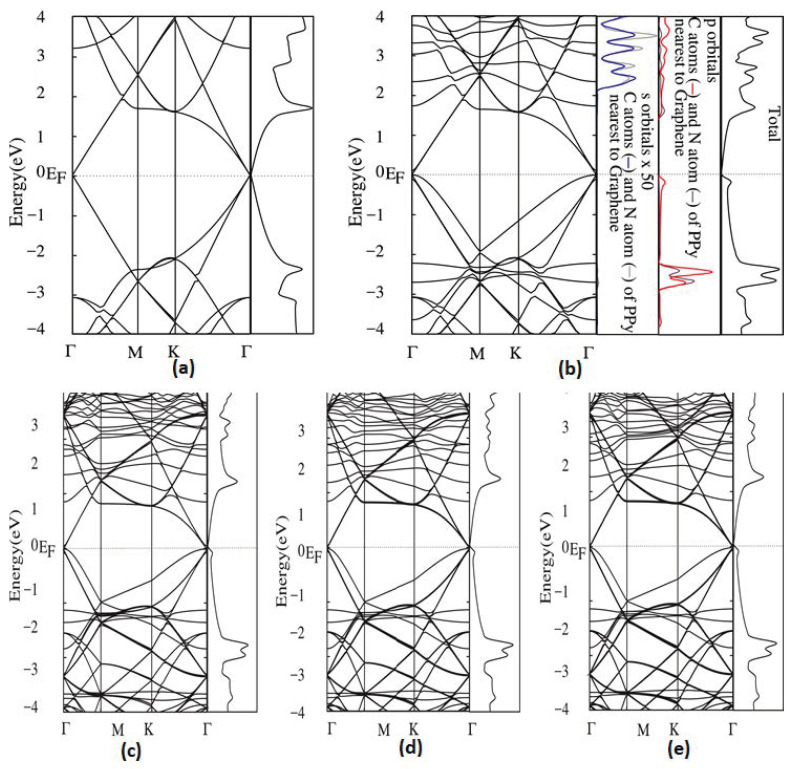
Electronic band structure of (**a**) pristine graphene (3 × 3) and (**b**) PPy on graphene at B site and 60° orientation. Energies are relative to the Fermi level. Right panels show the density of states (DOS) of (**a**,**b**). In (**b**) the PDOS, the projected density of states on the PPy heavy atoms are shown for the s-orbitals in blue for C and grey for N in the first panel, while the p-orbitals are depicted in the central panel in red for C and grey for N. (For interpretation of the references to color in this figure legend, the reader is referred to the web version of this article.) Electronic band structures of PPy confined between two graphene layers separated at different distances: (**c**) 8 Å, (**d**) 11 Å, (**e**) 14 Å. Right-most panels show the corresponding density of states. Energies are relative to the Fermi level. Reproduced with permission from [50]. Copyright 2018, Elsevier Science Ltd.

**Table 1 polymers-14-00704-t001:** Calculated atoms exchange energies using the Hartree–Fock [81], the OPM 48, 131, the LDA [80], and/or the gradient corrected functional by Becke [80] (in a.u.). Reproduced with permission from [82]. Copyright 1998, Elsevier Science Ltd.

Atom	HF	Optimized Potential Method (OPM)	LDA	B
He	−1.026	−1.026	−0.884	−1.025
Be	−2.667	−2.666	−2.312	−2.658
Ne	−12.108	−12.105	−11.03	−12.14
Mg	−15.994	−15.988	−14.61	−16.00
Ar	−30.185	−30.175	−27.86	−30.15

**Table 2 polymers-14-00704-t002:** Atoms calculated total energies by the Hartree–Fock; the OPM, the LDA, and GGA techniques (in a.u.). Reproduced with permission from [82]. Copyright 1998, Elsevier Science Ltd.

Atom	Exp.	HF	OPM	LDA	GGA
He	−2.9037	−2.8617		−2.975	−2.8989
C	−37.8450	−37.6886	−37.6865	−38.0522	−37.8243
Ne	−128.939	−128.547	−128.546	−129.317	−128.945
Si	−289.383	−288.854	−288.850	−289.912	−289.368
Cl	−460.217	−459.482	−459.477	−460.838	−460.162

**Table 3 polymers-14-00704-t003:** Density functional, conventional quantum chemical correlation energies (QC) and the variation between them [84] (in a.u.). Reproduced with permission from [82]. Copyright, 1998, Elsevier Science Ltd.

	*DFT*	*QC*	Δ *(*|*E*_*c,exact*_^*QC*^ − *E*_*c,exact*_^*DFT*^|/*E*_*c,exact*_^*DFT*^)	Δ% ((|*E*_*c,exact*_^*QC*^ − *E*_*c,exact*_^*DFT*^|/*E*_*c,exact*_^*DFT*^)100)
H^−^	−0.041995	−0.039821	+0.002174	5.20
He	−0.042107	−0.042044	+0.000063	0.20
Be^+2^	−0.044274	−0.044267	+0.000007	0.02
Ne^+8^	−0.045694	−0.045693	+0.000001	0.002
Be	−0.096200	−0.094300	+0.001900	2.00
Ne	−0.394000	−0.390000	+0.004000	1.00

**Table 4 polymers-14-00704-t004:** Correlation energies of atoms obtained by diverse approximate correlation energy functionals (in a.u.) [88]. Reproduced with permission from [82]. Copyright 1998, Elsevier Science Ltd.

	WL	LYP	GGA	LW	LDA	Exp
He	0.042	0.043	0.046	0.042	0.112	0.042
Be	0.094	0.094	0.094	0.094	0.223	0.094
Ne	0.383	0.383	0.383	0.374	0.743	0.392
Mg	0.444	0.459	0.451	0.462	0.888	0.444
Ar	0.788	0.750	0.771	0.771	1.426	0.787
Kr	1.909	1.748	1.916	1.948	3.267	
Xe	3.156	2.742	3.150	3.174	5.173	
Li^+^	0.044	0.047	0.051	0.060	0.134	0.044
Be^2+^	0.045	0.049	0.053	0.075	0.150	0.044
Ne^6+^	0.109	0.129	0.123	0.187	0.334	0.187
B^+^	0.101	0.106	0.103	0.114	0.252	0.111
Li^−^	0.0805	0.0732	0.078	0.069	0.182	0.073
F^−^	0.368	0.362	0.362	0.332	0.696	0.400

**Table 5 polymers-14-00704-t005:** Correlation energies of molecules obtained by various model correlation energy functionals. Reproduced with permission from [82]. Copyright 1998, Elsevier Science Ltd.

Molecule	WL	LYP	LW	PW	Exp
H_2_	0.049	0.038	0.029	0.046	0.041
Li_2_	0.136	0.133	0.134	0.137	0.122
Be_2_	0.231	0.200	0.193	0.205	0.205
B_2_	0.336	0.289	0.265	0.296	0.330
C_2_	0.446	0.384	0.344	0.391	0.514
N_2_	0.532	0.483	0.435	0.490	0.546
O_2_	0.621	0.583	0.533	0.588	0.657
F_2_	0.683	0.675	0.633	0.671	0.746
H_2_O	0.386	0.340	0.314	0.347	0.367
NH_3_	0.376	0.318	0.268	0.338	0.338
CH_4_	0.369	0.294	0.241	0.320	0.293
HF	0.377	0.363	0.335	0.367	0.387
LiH	0.088	0.089	0.083	0.092	0.083
LiF	0.417	0.418	0.343	0.415	0.447
HCN	0.525	0.464	0.410	0.478	0.527
CO	0.516	0.484	0.440	0.488	0.550
H_2_O_2_	0.690	0.638	0.569	0.652	0.691
C_2_H_2_	0.504	0.443	0.386	0.466	0.476
C_2_H_6_	0.678	0.551	0.426	0.577	0.553
C_2_H_4_	0.593	0.497	0.417	0.529	0.528
CO_2_	0.865	0.791	0.720	0.807	0.829

**Table 6 polymers-14-00704-t006:** Atomization energies (kcal mol^−1^) for some molecules estimated with DFT’s HF, MP2, LDA, BLYP, adopting 6-31G* basis [91] and BP, BRP, by use of 6-31G basis [92]. Reproduced with permission from [82]. Copyright, 1998, Elsevier Science Ltd.

	HF	MP2	LDA	BLYP	BP	BRP	Exp
H_2_	75.9	86.6	100.2	103.2	107.8	106.9	103.3
LiH	30.4	39.8	57.5	54.9	55.8	58.9	56.0
NH_3_	170.2	232.4	306.0	270.1	289.5	286.7	276.7
C_2_H_2_	271.9	365.6	438.6	383.4	398.6	404.0	388.9
H_2_CO	237.8	335.5	417.6	361.8	371.5	372.5	357.2
F_2_	−34.3	36.8	83.6	54.4	49.6	47.1	36.9

**Table 7 polymers-14-00704-t007:** Bond length (Å) for selected first-row diatomic molecules determined using HF, MP2, LDA, BLYP, adopting 6-31G* basis [91] and BP, BRP, by use of 6-31G basis [92]. Reproduced with permission from [82]. Copyright 1998, Elsevier Science Ltd.

	HF	MP2	LDA	BLYP	BP	BRP	Exp
H_2_	0.730	0.738	0.765	0.748	0.747	0.741	0.741
BeH	1.348	1.348	1.370	1.355	1.356	1.353	1.343
LiF	1.555	1.567	1.544	1.561	1.580	1.582	1.564
CO	1.114	1.150	1.142	1.150	1.135	1.130	1.128
N_2_	1.078	1.130	1.111	1.118	1.103	1.101	1.098
NO	1.127	1.143	1.161	1.176	1.160	1.158	1.151

## Data Availability

The data presented in this study are available on request from the corresponding author.

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
