# Peer review of "Prospect of DFT Utilization in Polymer-Graphene Composites for Electromagnetic Interference Shielding Application: A Review"

_polymers, 2022, doi:10.3390/polym14040704_

Round 1
Reviewer 1 Report
In this manuscript entitled “Utilization of DFT in Polymer-Graphene Composites for Electromagnetic Interference Shielding Application: A Review”, the authors reviewed density functional theory based method to model infinite systems of graphene-based polymer composites or their hybrids. The topic and results seem interesting and meaningful. I would like to support its publication on Polymers. However, there are some points could be addressed before its acceptance for publication.
- According to the Bloch’s theorem, a finite model with periodical boundary condition is necessary. Hence, how to model the polymer composites in periodical boundary condition should be introduced and discussed.
- For a technical review, the introduction of currently widely used density functional theory codes is beneficial for the readers.
- To understand the graphene-based composites, the materials in the graphite-C3N4 family must be mentioned, which should be of general interest.
Author Response
Response #1. Thank you for your valuable comments/suggestions. Modelling of the polymer composites in periodical boundary conditions as per available literature have been added in the revised manuscript. Within DFT in principle, it is probable working out linearly scaled schemes for molecules and their composites using the Kohn–Sham model like N3. (Line 660-681).
Response #2. The widely used density functional theory codes have been added in the introduction portion of the revised manuscript as per your comment. (Line 290-291).
Response #2. Thank you for your suggestive comment: seeing this review article is non-exhaustive, we have introduced graphite-C3N4 family materials in the revised manuscript. In future discussions in this regard, we will elaborate more on these materials as per your suggestion. (Line 456-460).
Reviewer 2 Report
Dear Authors
The submitted manuscript describes the concepts needed to model infinite systems of graphene-based polymer composites or their hybrids. The manuscript presented concerns an interesting and actual subject. This manuscript can be accepted after major revision. The following suggestion and comments should be taken:
- The overall English needs to be improved. Please seek guidance from a native English speaker if possible (commas, plural form, "the" "a", and others could be corrected).
- Please add in the introduction information about DFT calculations for other carbons for enhancement. Please cite (1) Physica B: Condensed Matter 541, (2018), 6-13, https://doi.org/10.1016/j.physb.2018.04.023 (2) Bull Mater Sci 41, 76 (2018). https://doi.org/10.1007/s12034-018-1603-5 (3) Surface Science 711, 121876, (2021) https://doi.org/10.1016/j.susc.2021.121876
- Why authors do not describe about carbon ends saturated with hydrogen atoms in carbon materials? Please explain in the comments. The dangling bonds of carbon ends may affect the structural and electronic properties of edge states.
- It is suggested to describes 2-6 sentences about DFT calculations for different size graphene systems to investigate the dependence on structure size.
- Please add information about basis set and their influence on final results.
- Authors are suggested to describe some future plans in conclusions to enhancement this review.
Author Response
Response #1. The English language has been revised as per your valuable comment.
Response #2. Information about DFT calculations for other carbons for enhancement have been added in the introduction in the revised manuscript. (Line 461-483).
Response #3. Thank you for your suggestive comment. Seeing the review is non-exhaustive, we believe this information will be added in our future write-up.
Response #4. The information about different size graphene systems for investigating its dependence on structure size has been added in the revised manuscript. (Line 75-483).
Response #5. The information about basis set and their influence on final results have been added in the revised manuscript. (Line 138--164).
Response #6. Some future plans in conclusions towards the enhancement of this review has been added to the conclusion portion of the revised manuscript. (Line 880-882).
Reviewer 3 Report
The manuscript "Utilization of DFT in Polymer - Graphene Composites for Electromagnetic Interference Shielding Applications: A Review" is an interesting work that meet the journal's rigors and I recommend its publication. I made the following comments:
- Some phrases from Abstract( line 10-14) are repeat in Introduction(line 61-63):
- The objectives of the work are not clearly specified;
- The originality elements towards other works on the same topic should be mentioned;
- There are many typing errors;
- A higher level of content systematization would facilitate the text reading
Author Response
Response #1. Thank you for your valuable observative comments. The abstract has been revised as per your comment.
Response #2. This review focuses on projecting the necessity of DFT utilization in polymer-graphene composites systems design and fabrication for EMI shielding application. This statement has been added to the revised manuscript. (Line 110-114).
Response #3. The originality elements towards other works on the same topic have been mentioned in the revised manuscript. Particularly, with reference to other niches innovative utilization of DFT for novel graphene-based polymeric materials as per available literature, the key element of originality of the present article is to have projectively pulled out the bright prospect DFT utilization in the design/engineering of novel graphene-based polymeric materials for EMI shielding application and other technological advancement intensity as well as relevance. (Line 134-137).
Response #4. The typographical errors have been corrected as per your suggestions.
Response #5. Your valuable suggestion with regards to higher level of content systematization have been applied in the revised manuscript.
Round 2
Reviewer 1 Report
The manuscript can be accepted for publication as is.
Reviewer 2 Report
Accept in present form